# High fat diet induces obesity, alters eating pattern and disrupts corticosterone circadian rhythms in female ICR mice

Kelsey Teeple[1], Prabha Rajput[1,2], Maria Gonzalez[1], Yu Han-Hallett[3], Esteban Fernández-Juricic[4], Theresa Casey[1] *

1 Department of Animal Sciences, Purdue University, West Lafayette, Indiana, United States of America, 2 Neurotherapeutics Lab, Pharmaceutical Engineering and Technology, Indian Institute of Technology (Banaras Hindu University), Varanasi, India, 3 Bindley Bioscience Center, Purdue University, West Lafayette, Indiana, United States of America, 4 Department of Biological Sciences, Purdue University, West Lafayette, Indiana, United States of America

* theresa-casey@purdue.edu

**Data Availability Statement:** All data is available through Figshare by this link: https://figshare.com/s/7ad3a3b95ce66b1575df.

## Abstract

Circadian, metabolic, and reproductive systems are inter-regulated. Excessive fatness and circadian disruption alter normal physiology and the endocrine milieu, including cortisol, the primary stress hormone. Our aim was to determine the effect feeding a high fat diet to female ICR mice had on diurnal feeding pattern, weight gain, body composition, hair corticosterone levels and circadian patterns of fecal corticosterone. Prepubertal (~35d of age) ICR mice were assigned to control (CON; 10% fat) or high fat (HF; 60% fat) diet and fed for 4 wk to achieve obesity under 12h light and 12h of dark. Feed intake was measured twice daily to determine diurnal intake. Mice were weighed weekly. After 4 wk on diets hair was collected to measure corticosterone, crown-rump length was measured to calculate body mass index (BMI), and body composition was measured with EchoMRI to determine percent fat. HF mice weighed more (P<0.05) after week two, BMI and percent body fat was greater (P<0.05) in HF than CON at the end of wk 4. HF mice consumed more during the day (P<0.05) than CON mice after 1 week on diets. Hair corticosterone was higher in HF mice than in CON (P<0.05). Fecal circadian sampling over 48hr demonstrated that HF mice had elevated basal corticosterone, attenuated circadian rhythms, and a shift in amplitude. High fat feeding for diet induced obesity alters circadian eating pattern and corticosterone rhythms, indicating a need to consider the impact of circadian system disruption on reproductive competence.

## Introduction

The rate of obesity has risen dramatically over the last decades. The most recent data show a continued rise in obesity among women of childbearing age [1, 2]. Prepregnancy obesity is linked to decreased fertility and adverse health outcomes such as gestational diabetes, hypertension, preeclampsia, preterm delivery, and increased neonatal morbidity and mortality

**Funding:** This activity was funded by Purdue University as part of AgSEED Crossroads funding to support Indiana's Agriculture and Rural Development. The recipient of the award is TC.

**Competing interests:** The authors have declared that no competing interests exist.

[2, 3]. Metabolism and circadian clocks are intimately associated and reciprocally regulate each other. Obesity has been linked to the dysregulation of the endocrine system and circadian rhythm disruption. In reciprocal, circadian rhythm disruption has been linked to lower fertility and poorer outcomes of maternal-fetal health as well as lactation [4–6].

The circadian timing system functions to coordinate the timing of physiology and behavior and synchronize these to the outside world. Circadian clocks generate 24 h rhythms that are predictive of regularly occurring environmental cues, such as light-dark cycles and food availability [7]. In mammals, the regulation of clocks is hierarchical, with the master clock residing in the suprachiasmatic nuclei (SCN) of the hypothalamus. Temporal information is integrated in the SCN and relayed to the peripheral clocks located in other organs through output rhythms that include circadian rhythms of circulating hormones such as cortisol to coordinate the timing of physiological processes across the body [8]. Cortisol rhythms naturally peak right before the onset of waking in many species and then dwindle throughout the day [9, 10]. Cortisol is also released in response to stress, and the rise of cortisol prior to waking is thought to help prepare for the typical stresses that may be encountered throughout the day [10, 11]. Chronic stress and obesity are associated with the diminishment of robust cortisol circadian rhythms [12].

Chronic circadian rhythm disruption as experienced by night shift workers can negatively affect health, with consequences including decreased fertility, metabolic disorder, and obesity [13, 14]. The metabolic disturbances associated with circadian disruption result from the highly integrated nature of these systems. Mice homozygous for the *Clock-Δ19* mutation, which disrupts circadian rhythms, exhibit excess energy intake and altered patterns of eating that resulted in weight gain. Male *Clock-Δ19* mice develop hyperglycemia, hypoinsulinemia and altered lipid metabolism, as evident by hypertriglyceridemia and glycogen accumulation in hepatocytes, relative to wild-type controls [15]. Hyperglycemia, hypoinsulinemia, hypertriglyceridemia, and glycogen accumulation is classic metabolic syndrome. Male C57BL/6J mice fed a high fat diet consumed significantly more during the light phase when mice are typically less active, demonstrating high fat diets alone dampen behavioral rhythms even before significant weight gain occurs. Altered eating behavior of mice on high fat diet was accompanied by an attenuation of core circadian genes' expression rhythms in the hypothalamus, liver, and epididymal fat [16, 17].

An understanding of how obesity is linked to circadian disruption may be important to understanding the link between obesity and reduced fertility, compromised maternal-offspring health during gestation, and the negative impacts of excess fat on lactation competence. Rodent models are frequently used to investigate underlying biology that affects human health and attempting to unravel the effects of obesity is no exception. To induce obesity in rodents, hypercaloric diets with excessive amounts of fat are typically used to cause diet-induced obesity (DIO) [18, 19]. Chronic overexposure to hypercaloric food increases adipose accumulation and causes weight gain [20]. There are a number of factors that influence the response to DIO, including the strain of mice, gender, genetic and environmental factors [18, 21, 22]. Inbred mice are commonly used in mouse models, however the homozygosity of inbred mice does not accurately capture the diversity of the human population like an outbred strain can [18]. Here we used ICR mice, an outbred strain that originates from Switzerland and is renowned for their excellent maternal characteristics, to determine the effect of feeding a high fat diet on diurnal eating patterns, weight gain, body composition, hair corticosterone levels and circadian rhythms of fecal corticosterone. We predicted that animals assigned to the high fat diet would have altered eating pattern increased levels of corticosterone and an attenuation of rhythms, increased weight gain, and a higher percentage of body fat that those on the control diet.

## Materials and methods

### Animals and diets

Before beginning this study, animal use protocols were reviewed and approved by Purdue University's Institutional Animal Care and Use Committee (protocol # 2104002135). Three-week-old female ICR mice (n = 87; CD1, Envigo, Indianapolis, IN, USA) were ear tagged upon arrival and allowed to acclimate for two weeks. Experiments were run with two cohorts (cohort 1 n = 42, cohort 2 n = 43) of mice. After two weeks of acclimation, mice were randomly assigned into one of two dietary treatments: control (CON, n = 36) and high fat (HF, n = 49) diet. Animals were part of a larger study aimed at understanding the effect of high fat diet induced prepregnancy obesity on pregnancy and lactation. The larger cohort of females assigned to HF diet accounted for an expected decreased reproductive efficiency in this group (80% of animals on CON diet) [23]. Animals were weighed on day of diet assignment, and no difference in initial body weight was found between the groups (CON = 21.7 g ± 0.52 g, HF = 21.7 g ± 0.45 g). Mice were housed in groups of 3–5 per cage based on diet and fed for *ad libitum* intake of either control (CON; Research Diets #D12450J, 10% of total kcal energy is fat and 7% of total kcal is sucrose; 3.85 kcal/g) or high fat (HF, Research Diets #D12492, 60% of kcal energy is fat and 7% of kcal energy is sucrose; 5.24 kcal/g) diets, that were matched based on sucrose content and isonitrogenous. The fat composition of the control diet was 23.70 g of soybean oil per 1000 g of diet and 18.96 g lard per 1000 g diet, and the high fat diet was composed of 32.31 g of soybean oil per 1000 g of diet and 316.60 g of lard per 1000 g of diet. Comprehensive components are available at Research Diets, Inc and in our previous publication [24]. Mice remained on the respective diets for the entirety of the study and were exposed to 12 h light- 12 h dark cycles, with lights on at 0600 and off at 1800.

Mice were weighed once a week. At the end of week four, females in cohort 1 (CON n = 18, HF n = 23) were weighed and anesthetized using 3% isoflurane gas at a rate of 1.0 L/minute oxygen. While under anesthesia, crown-rump length was measured and a portion of hair was shaved using Wahl Mini Pro corded trimmers (Wahl Clipper Corporation, Sterling, IL, USA) for down-stream analysis of corticosterone content. Body mass index (BMI) was calculated for cohort 1 as the ratio of weight to crown-rump length. Alternatively, after four weeks on experimental diets, mice in cohort 2 (CON n = 18, HF n = 20) had their body composition (percent fat) analyzed using EchoMRI 700 (EchoMRI, Houston, TX, USA). The EchoMRI emits radio pulses that results in hydrogen proton spins releasing radio signals that differentiates fat, lean, and free water based on specific radio pulse signals. The EchoMRI was calibrated with canola oil prior to use.

Over the two cohorts, five HF mice and one CON mouse were euthanized at the end of the four-week period due to weight loss that was indicative of morbidity. All group housed data (feed intake and fecal cortisol) included these mice, individual data (weight, BMI, percent body fat, hair corticosterone) were analyzed with and without these mice. Moribund mice were euthanized via $CO_2$ inhalation at a fill rate of 30% chamber volume per minute for 3 minutes followed by cervical dislocation as a secondary method. $CO_2$ inhalation is an approved method of euthanasia under the American Veterinary Medical Associate to alleviate suffering. At the conclusion of the larger study, all mice were euthanized via $CO_2$ inhalation followed by cervical dislocation.

### Determination of diurnal feed intake

Diurnal feed intake was calculated by determining the amount consumed during the light (0600–1800) or dark (1800–0600) phase of the light-dark cycle on a per cage basis. Feed was

weighed Monday-Friday at 0600 and 1745. To determine what was consumed during the light phase, the amount of food that was weighed at 1745 was subtracted from the food that was weighed at 0600 for the same day. To determine what was consumed during the dark phase, food that was weighed at the next day's 0600 timepoint was subtracted from the previous day's 1745 measure. To correct for varying numbers of animals per cage, intake was divided by the number of animals in each cage and expressed as grams (g) consumed per mouse per cage. Kcal intake was calculated by multiplying $g*3.85$ kcal/g for CON, and $g*5.24$ kcal/g for HF diet.

## Hair corticosterone extraction and analysis using liquid chromatography-tandem mass spectrometry (LC-MS/MS)

Extraction and LC-MS/MS were performed in Purdue University's Metabolite Profiling Facility in Bindley Life Sciences. Approximately 30 mg of hair was placed in a 15 ml conical, 2 mL of isopropyl alcohol was added, swirled for 15 seconds, and then alcohol was decanted. Cleaned hair was wrapped in aluminum foil and dried in a warm room at 32°C until all the isopropyl alcohol evaporated. Cleaned hair was placed into a Precellys MK28 (Bertin Technologies SAS, Montigny-le-Bretonneux, France) lysing tube and weighed. Samples were loaded into the Precellys homogenizer centrifuge and set to 6500 rpm, 3 cycles, with the length of cycle rotation set to 30 seconds, and the length of rest set to 20 seconds. After homogenizing, 1 mL of methanol + ISTD (10 ng deuterated d8-corticosterone, Toronto Research Chemicals, Toronto, ON, Canada) was added to the samples and then incubated on a rocker overnight at 4°C. After incubation, samples were centrifuged and the supernatant was collected and dried using a Savant SPD 2010 speed-vac (Waltham, MA, USA). Samples were reconstituted in 50 μL of Amplifex Keto Reagent (AB Sciex, Framingham, MA, USA), incubated at room temperature for one hour. After incubating, 30 μL of ddH2O was added and the samples were centrifuged at 13000 rpm for five minutes. The supernatant was transferred to an LC vial and analyzed using an Agilent 6470 QQQ LC-MS/MS system (Agilent, Santa Clara, CA, USA) with a C18 column and water/ACNN+ 0.1% FA buffer.

## Circadian fecal collection, corticosterone extraction and analysis using enzyme linked immunosorbent assay (ELISA)

During week four, each cage of females from cohort 1 (CON n = 4 cages, HF n = 5 cages) had fecal samples collected every 4 h over a 48 h period. During the dark phase, red light headlamps were worn for fecal collection. All the mice from one cage were placed in a new cage with fresh corn cob bedding, and the old bedding was transferred to a baggie, and fecal material was picked out and stored at—20°C. All fecal material were picked out from the bedding and weighed to determine amount produced over 4 h period.

Preliminary analysis of fecal extracts using LC-MS/MS indicated that amount could not readily be calculated because of interference of metabolites, and so fecal corticosterone was measured using a commercial ELISA kit. Corticosterone was extracted per Arbor Assay protocol. Approximately 0.2 g of fecal material was weighed and then crushed into a fine powder. One mL of ethanol was added per 0.1 g of fecal material and then vortexed for 30 s after resting for 2 min over the course of a 30 min period. Samples were then centrifuged at 4°C at 5000 rpm for 30 min. The supernatant was transferred to a new 2 mL microcentrifuge tube, and dried using a Savant speed-vac SC110 with a RVT400 refrigerated vapor trap (Thermo Fisher Scientific Inc., Waltham, MA, USA). Once samples were dried, they were stored at -20°C in a desiccator. Samples were reconstituted in absolute ethanol and analyzed using the corticosterone ELISA from Arbor Assays (catalog no. K014, Ann Arbor, MI, USA) following

manufacturer's protocol. Absorbance was read at 450 nm on the Spark 10M multimode microplate reader (Tecan Trading AG, Switzerland).

## Statistical analysis

All statistical analyses were performed using R version 4.1.2 [25]. All the data and code for the analyses are available on figshare: https://figshare.com/s/7ad3a3b95ce66b1575df. We considered a P-value significant when $< 0.05$.

We used generalized linear mixed models with the afex package in R to analyze three dependent variables: body weight, food consumption in grams, and food consumption in kcalories [26]. For the body weight analysis, we considered four fixed factors: diet, week, cohort, and the interaction between diet and week. Week was considered a within-subject factor as each mouse was measured every week. The identity of the mouse was considered a random factor in our model. For the food consumption in grams and in kcalories, we considered several fixed factors: diet, time of day, week, cohort, three two-way interactions (diet x time of day, diet x week, time of day x week), and one three-way interaction (diet x time of day x week). The number of mice per cage was included in the models as a potential confounding factor. Week and time of day were considered within-subject factors as each cage was measured at different weeks and time of day. The identity of the cage was considered a random factor in our model.

Our mixed model selection procedure followed Singmann and Kellen (2019), whereby we assessed random structures with different level of complexity (from more to less complex) until the models would converge [27]. We eventually chose the most complex random structures that would match the statistical results of less complex structures (i.e., random intercepts). For all models, assumption of normality of residuals and homogeneity of variance were checked. We checked for model assumptions, and performed log-transformations when necessary. We used the package emmeans to test for post-hoc comparisons that use the Tukey's method of P-value adjustment for multiple comparisons [28].

We analyzed the effects of diet on BMI (cohort 1), percent body fat (cohort 2), and hair corticosterone with general linear models. We checked for model assumptions (normality of residuals, homogeneity of variances). We also explored the potential association between body weight/BMI and hair corticosterone with Pearson's product moment correlations that were run within each diet treatment.

Fecal weight (g), corticosterone concentration (pg/mL), and total corticosterone (μg) data were analyzed for fit to one and two-component cosine curves. R version 4.1.2 and card version 0.1.0 was used for the cosinor analysis [29]. The mean values at each time point across all cages within diet treatments were used for fit analysis. For both one and two component analyses, a period of 24 h was used. The F statistic and $R^2$ for one and two-component cosinor fitted curves were estimated with the population means and fitted values based on the equations described by Cornelissen et al [30]. Finally, we compared the $R^2$ values of the one and two-component curves and chose the best fit for each variable.

## Results

### Impact of diet on body weight, BMI, and percent body fat

We found that body weight was significantly influenced by diet, week, and their interaction, but without significant differences between cohorts (Table 1). Mice on HF diet weighed more ($26.60 \pm 0.39$ g) than those on CON diet ($24.40 \pm 0.46$ g). As expected, weight increased over time (week 0, $21.60 \pm 0.36$ g; week 1, $24.30 \pm 0.33$ g; week 2, $26.20 \pm 0.32$ g; week 3, $27.40 \pm 0.33$ g; week 4, $28.0 \pm 0.37$ g). There was a significant interaction between diet and

**Table 1. Results from general linear mixed models on body weight, food consumed (grams), and kcal consumed.**

|  | F | d.f. | P |
|---|---|---|---|
| *Weight* |  |  |  |
| Diet | 12.75 | 1, 82 | **< 0.001** |
| Week | 154.99 | 4, 139.76 | **< 0.001** |
| Cohort | 0.40 | 1, 82 | 0.529 |
| Diet x Week | 13.77 | 4, 139.76 | **< 0.001** |
| *Food consumed (grams per day)* |  |  |  |
| Diet | 20.53 | 1, 13.9 | **< 0.001** |
| Time of day | 894.58 | 1, 16 | **< 0.001** |
| Week | 1.08 | 3, 27.2 | 0.375 |
| Cohort | 2.27 | 1, 14.11 | 0.154 |
| Mice per cage | 49.20 | 1, 14.3 | **< 0.001** |
| Diet x time of day | 33.89 | 1, 16 | **< 0.001** |
| Diet x week | 0.92 | 3, 27.2 | 0.443 |
| Time of day x week | 3.67 | 3, 27 | **0.024** |
| Diet x time of day x week | 6.03 | 3, 27 | **0.003** |
| *Kcal consumed* |  |  |  |
| Diet | 46.54 | 1, 14 | **< 0.001** |
| Time of day | 360.79 | 1, 16 | **< 0.001** |
| Week | 5.19 | 3, 26.4 | **0.006** |
| Cohort | 0.14 | 1, 14 | 0.711 |
| Mice per cage | 32.61 | 1, 14.1 | **< 0.001** |
| Diet x time of day | 0.43 | 1, 16 | 0.520 |
| Diet x week | 0.20 | 3, 26.4 | 0.897 |
| Time of day x week | 5.01 | 3, 25.9 | **0.007** |
| Diet x time of day x week | 3.16 | 3, 25.9 | **0.042** |

Significant results (P < 0.05) are marked in bold.

week, which overall showed the difference between HF and CON diets increased over time (Fig 1A). At the time of assignment to experimental diets, there was no difference in the weight between HF and CON mice (t 144 = 0.24, P = 0.81; Fig 1A). The difference in weight between diet treatments was significant in each of the successive weeks.

At the end of the experiment (week 4), HF mice had significantly higher BMI than CON mice (F 1,40 = 11.03, P = 0.002) (Fig 1B). Additionally, HF mice had significantly higher percent of body fat than CON mice (F 1,41 = 10.41, P < 0.002) (Fig 1C).

## Impact of diet on feeding behavior

Food consumed during the day (in grams) was significantly affected by diet, time of the day, the two-way interaction between diet and time of the day and between time of the day and week, and the three-way interaction among diet, time of the day and week (Table 1). Food consumption was higher in HF than in CON diet. In week 1, there was no significant difference in food consumed between CON and HF diets during the day (t 73 = -1.96, P = 0.054) or night (t 86 = 0.15, P = 0.88; Fig 2A). In week 2, HF mice consumed significantly more food than those on CON diet during the day (t 74 = -7.19, P < 0.001), but during the night this pattern reversed, with CON mice consuming significantly more food than those HF (t 74 = 2.57, P = 0.012; Fig 2A). In weeks 3 and 4, mice on HF diet consumed significantly more food than

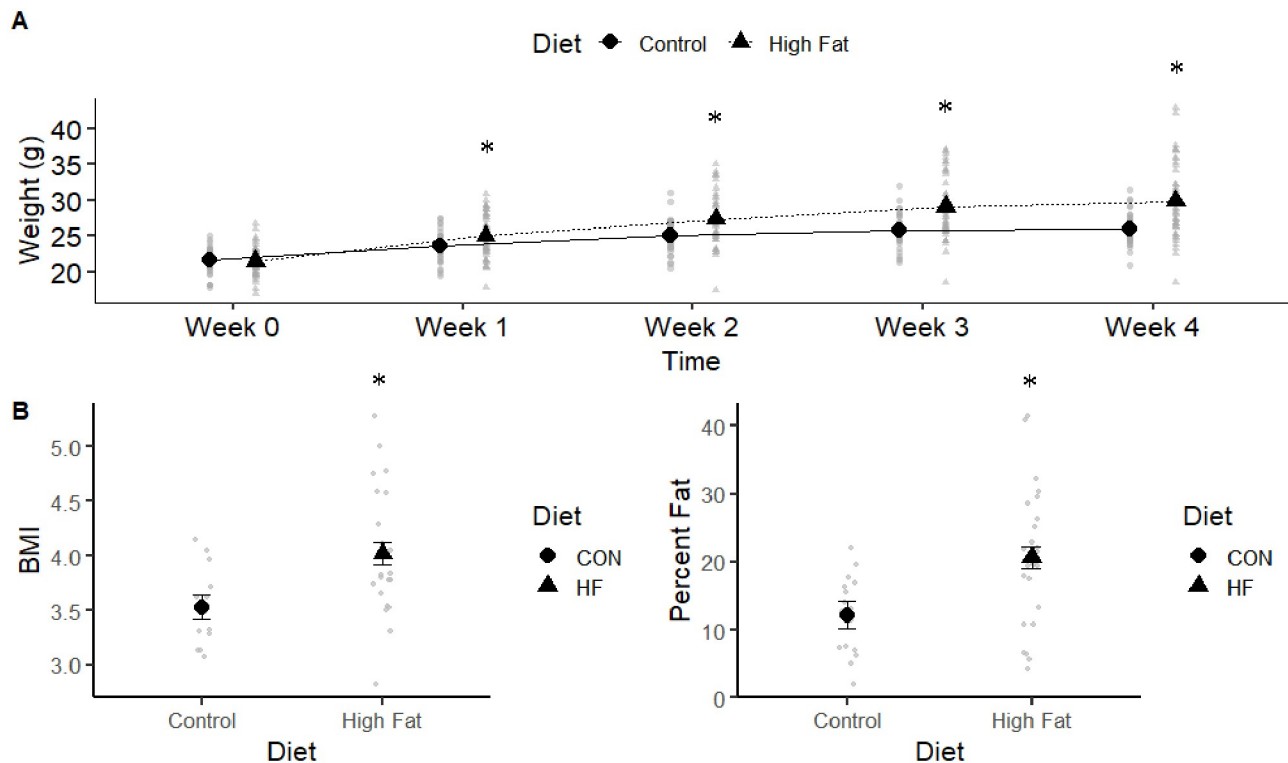

**Fig 1. Effect of diet on body weight and composition over four weeks.** A) Average weight by week, from the onset of experimental diets until 4 weeks across both cohort 1 and 2, CON (n = 36) and HF (n = 49). On week one and after, CON and HF weights are different (P<0.05), as determined by mixed model and post-hoc analysis. B) BMI measurements (ratio of weight to length) at the end of the four-week period, cohort 1 mice, CON (n = 18) and HF (n = 24). C) Percent fat, as measured by EchoMRI in cohort 2, CON (n = 18) and HF (n = 25). * indicates significant results (P < 0.05).

CON during the day (week 3, t 73 = -4.14, P < 0.001; week 4, t 67 = -5.40, P < 0.001), but there was no significant difference in food consumption between diets at night (week 3, t 73 = 1.57, P = 0.121; week 4, t 67 = 1.41, P = 0.162).

The amount of kcal consumed was significantly affected by diet, week, time of day, the two-way interaction between time of the day and week, and the three-way interaction among diet, time of the day and week (Table 1). HF mice (33.00 ± 0.80) consumed more calories than CON (24.60 ± 0.90). Mice consumed more calories during the night (46.70 ± 1.11) than during the day (10.80 ± 1.11). Calorie consumption decreased from weeks 1 (30.20 ± 0.87) and 2 (30.10 ± 0.81) to weeks 3 (27.00 ± 1.02) and 4 (27.70 ± 1.08). In week 1, there was no significant difference between diets in calorie consumption during the day (t 57.4 = -1.17, P = 0.247), but at night mice on HF diet consumed significantly more calories than CON mice (t 64 = -4.81, P < 0.001; Fig 2B). In week 2, mice on HF diet consumed significantly more calories than those on CON diet both during the day (t 51.4 = -3.57, P < 0.001) and at night (t 51.4 = -2.74, P = 0.009; Fig 2B). In week 3, there was no significant difference in calorie consumption between diets during the day (t 61.1 = -1.74, P = 0.086), but at night calorie consumption was higher in HF than in CON diet (t 61.1 = -2.58, P = 0.012; Fig 2B). Lastly, in week 4, there was no significant difference in calorie consumption during the night (t 68.2 = -1.89, P = 0.063), but mice in the high fat diet consumed more calories than those in the control diet during the day (t 68.2 = -2.43, P = 0.018 (Fig 2B).

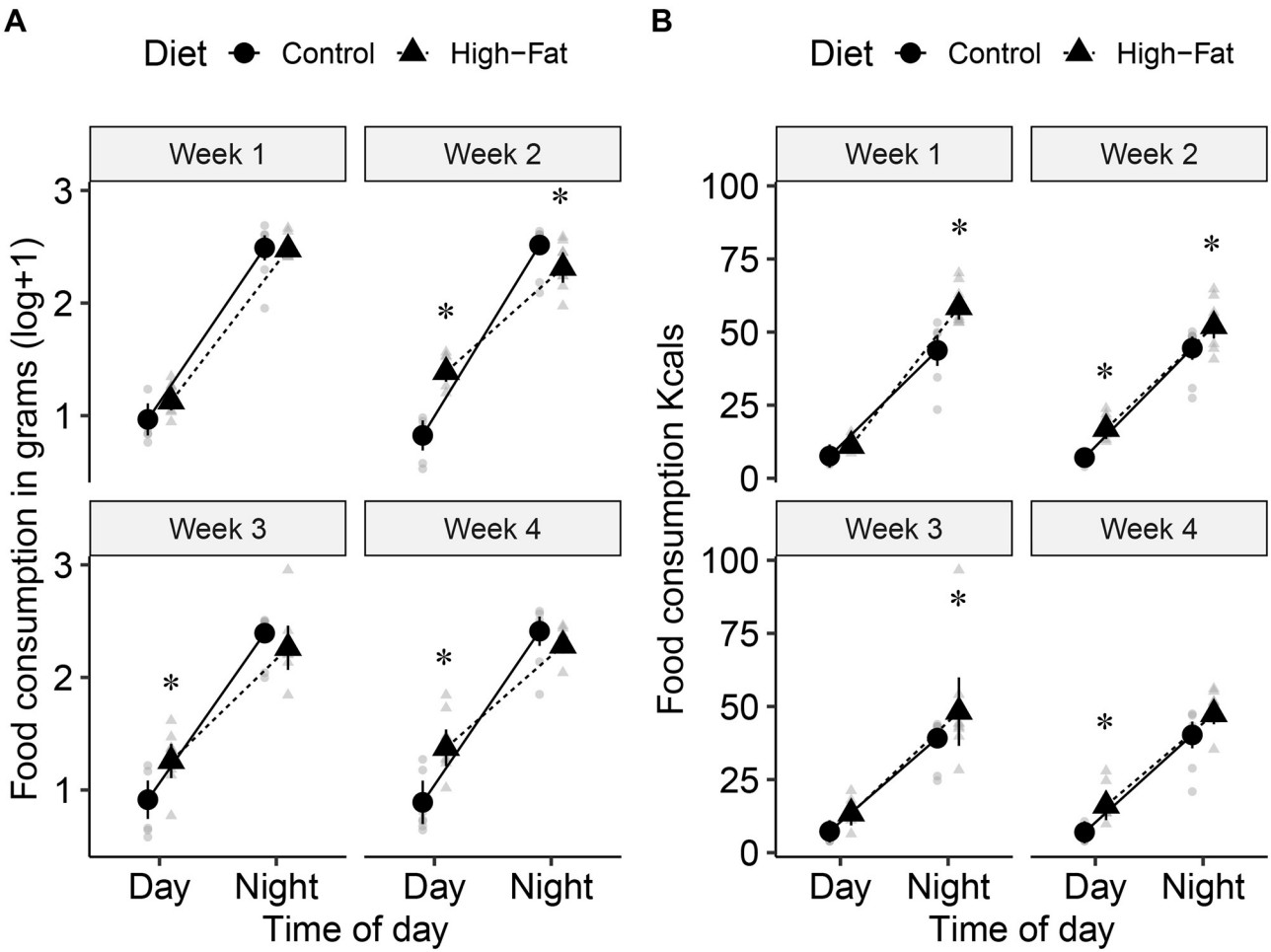

**Fig 2. Effect of diet on diurnal feeding behavior.** A) Weekly feed intake in grams (log + 1) comparing intake during day and night of CON (n = 8 cages) and HF (n = 10). B) Weekly feed intake in kcal during day and night of CON and HF. * indicates significant results (P < 0.05).

### Impact of diet on hair corticosterone levels

After four weeks on respective diets, HF mice had significantly greater levels of corticosterone in their hair than CON mice (F 1,39 = 9.53, P = 0.004) (Fig 3). To determine if there was any potential association between corticosterone and weight or BMI, Pearson's correlation analysis was performed within diets. Neither weight (Pearson's r = −0.35, P = 0.16) nor BMI (Pearson's r = 0.06, P = 0.80) were significantly associated with hair corticosterone in CON mice. Similarly, mice on HF diet, did not show a significant association between hair corticosterone and weight (Pearson's r = −0.08, P = 0.7) or BMI (Pearson's r = −0.21, P = 0.34). Moreover, Pearson's correlation analysis of hair corticosterone data across both treatments versus weight or BMI indicated there was no relationship between these variables.

### Cosine analysis of fecal weight and corticosterone

Fecal output and corticosterone data were analyzed for rhythmicity using single and two-component models. When comparing how well the data fit to 24 h cosine curves the $R^2$ was utilized. Corticosterone concentration better fit a two-component (CON $R^2$ = 0.47 and HF $R^2$ = 0.61) versus one component (CON $R^2$ = 0.43 and HF $R^2$ = 0.38) cosine curves (Fig 4A,

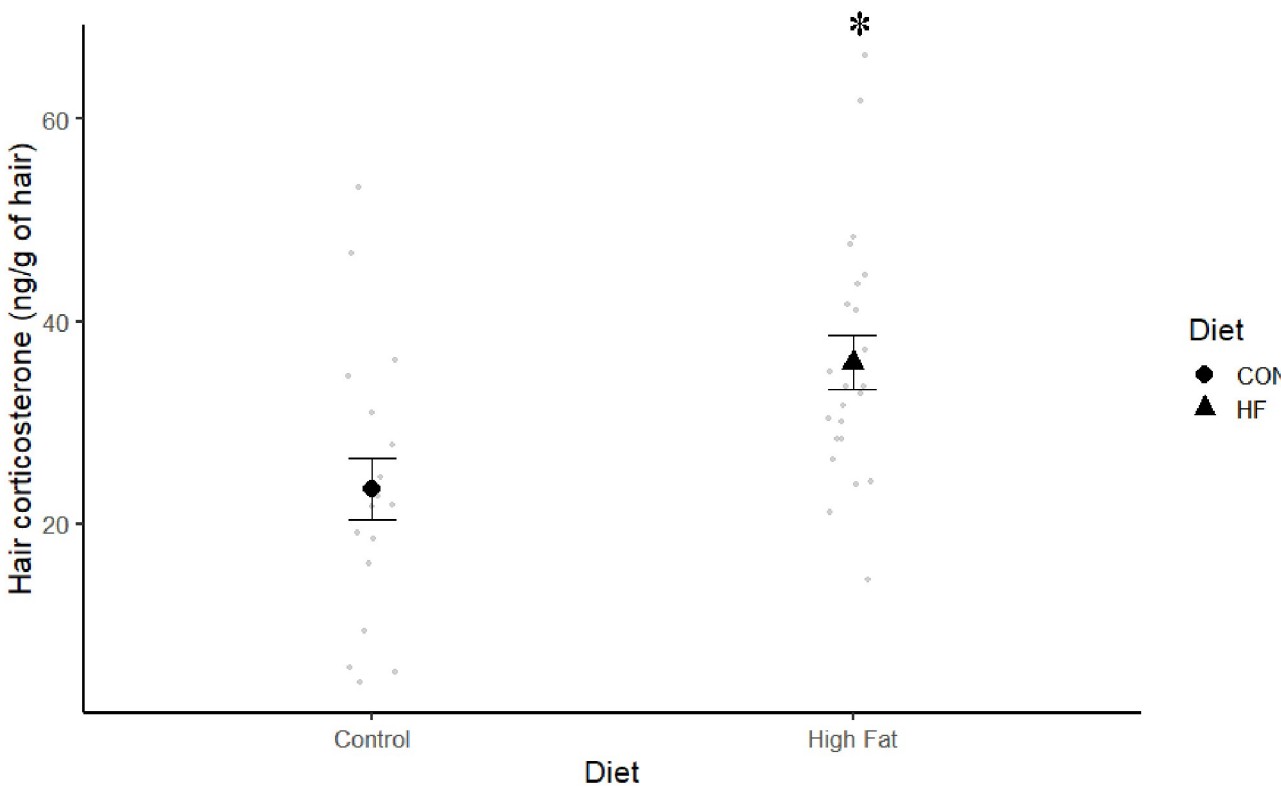

**Fig 3. Effect of diet on hair corticosterone levels in mice from cohort one between CON (n = 18) and HF (n = 23).** * indicates significant results (P < 0.05).

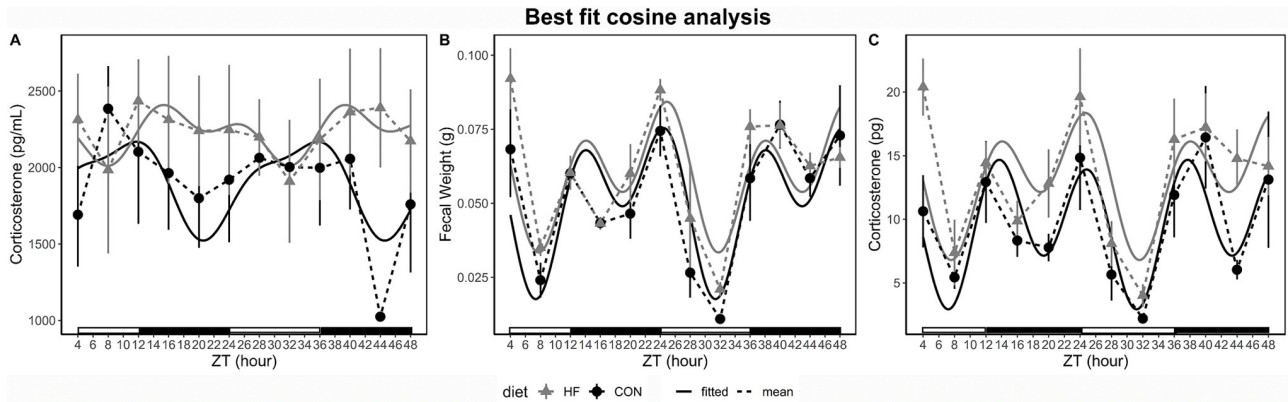

**Fig 4. Best fit cosine curve analysis for fecal corticosterone concentration, fecal weight, and total corticosterone produced.** A) Two-component cosine analysis of corticosterone concentration (pg/mL) over 48 hr. CON (n = 4 cages) $R^2$ = 0.47 and HF (n = 5 cages) $R^2$ = 0.61. B) Two-component cosine analysis of fecal weight (g) over 48 hr. CON (n = 4 cages) $R^2$ = 0.69 and HF (n = 5 cages) $R^2$ = 0.50. C) Two-component cosine analysis of total corticosterone in feces (pg) over 48 hr. CON (n = 4 cages) $R^2$ = 0.73 and HF (n = 5 cages) $R^2$ = 0.48. Black dash line represents mean of CON cages, and solid black line represents fitted cosine curve of CON cages. Gray dash line represents mean of HF cages, gray black line represents fitted curve of HF cages. Data points within dashed lines are mean ± standard deviation. White and black bars above the x-axis represent the light and dark phases of the light-dark cycle. ZT, refers to zeitgeber time, with ZT0 presenting the time of light onset on the day sampling was begun.

**Table 2. One-component and two-component cosine fit analysis of fecal corticosterone concentration, fecal weight, and total corticosterone secreted in feces.**

| One-component 24h period | | | | | | | |
|---|---|---|---|---|---|---|---|
| | Diet | MESOR | Amplitude | Acrophase | F value | P value | $R^2$ |
| Corticosterone concentration (pg/mL) | CON | 1897.27 ± 79.55 | 291.08 ± 122.51 | 9.33 | 3.35 | 0.08 | 0.43 |
| Fecal weight (g) | CON | 0.052 ± 0.006 | 0.016 ± 0.008 | 20.12 | 1.96 | 0.20 | 0.30 |
| Corticosterone secreted in feces (pg) | CON | 9.60 ± 1.30 | 2.14 ± 1.84 | 18.59 | 0.68 | 0.53 | 0.13 |
| Corticosterone concentration (pg/mL) | HF | 2235.96 ± 39.22 | 131.11 ± 55.47 | 18.24 | 2.79 | 0.11 | 0.38 |
| Fecal weight (g) | HF | 0.06 ± 0.006 | 0.012 ± 0.009 | 21.63 | 0.974 | 0.414 | 0.18 |
| Corticosterone secreted in feces (pg) | HF | 13.25 ± 1.46 | 2.91 ± 2.06 | 20.86 | 1.00 | 0.40 | 0.18 |
| Two-component 24h period | | | | | | | |
| | Diet | MESOR | Amplitude | Acrophase | F value | P value | $R^2$ |
| Corticosterone concentration (pg/mL) | CON | 1897.27 ± 86.61 | 291.08 ± 122.49 | 9.33 | 4.01 | 0.06 | 0.47 |
| Fecal weight (g) | CON | 0.052 ± 0.004 | 0.016 ± 0.006 | 20.12 | 10.04 | 0.005 | 0.69 |
| Corticosterone secreted in feces (pg) | CON | 9.60 ± 0.82 | 2.14 ± 1.16 | 18.59 | 12.20 | 0.003 | 0.73 |
| Corticosterone concentration (pg/mL) | HF | 2229.27 ± 35.46 | 131.11 ± 50.14 | 18.42 | 6.97 | 0.01 | 0.61 |
| Fecal weight (g) | HF | 0.06 ± 0.005 | 0.012 ± 0.008 | 21.63 | 4.52 | 0.044 | 0.50 |
| Corticosterone secreted in feces (pg) | HF | 13.25 ± 1.31 | 2.91 ± 1.86 | 20.86 | 4.20 | 0.052 | 0.48 |

Cosine curve analysis was performed in RStudio using cosinor2 package, where the population mean of diets was analyzed as a one component and two-component fit. The amplitude, mesor, acrophase (peak-time), and p-value were measured as the population mean for CON (n = 4 cages) and HF (n = 5 cages).

Table 2). The calculated time of the peak (acrophase) of corticosterone concentration was shifted from zeitgeber time (ZT) 9.33 h in CON cages, which was during the light phase, to 18.4 h ZT, which was in the dark phase of the light-dark cycle. The amplitude, or the distance from mesor to peak value, of corticosterone concentration was reduced in HF (131.1 ng/ml) versus CON (291.1 ng/ml) mice. Mesor, or the mean across the day, was elevated in HF mice (2229.3 ± 35.46 ng/ml) compared to CON (1897.3 ± 86.61 ng/ml). Daily variation of fecal weight and total fecal corticosterone fit two-component (CON $R^2$ = 0.69 and 0.73, HF $R^2$ = 0.50 and 0.48) cosine curves better than one-component (CON $R^2$ = 0.30 and 0.13, HF $R^2$ = 0.18 and 0.18) (Fig 4B and 4C).

## Discussion

Mice on this study were part of a larger experiment aimed at understanding the interaction among circadian, metabolic, and reproductive systems on reproductive competence during gestation and lactation. Data collected were from a four-week period that encompassed pubertal growth and development of female mice and ended with their full sexual maturation, when they are typically bred for studies of pregnancy and lactation [31]. Although there was a high degree of variability within each group, mice fed a high fat diet on average weighed more, had higher BMI and a higher percent body fat than those fed a control diet. Animals on the high fat diet also exhibited altered circadian feeding patterns, attenuated corticosterone circadian rhythms and significantly higher hair corticosterone levels, which is indicative of physiological stress and was found to be independent of their final body weight.

The greater weight and higher adiposity of mice on a high fat diet are consistent with previous work in our lab and others that have studied ICR mice [18, 19, 32]. After four weeks, the median weight of all mice was 27.5 g and 80% of mice above the median were HF mice. Prior to a significant difference in weight, HF mice displayed a disrupted eating behavior. In the second week of being on experimental diets, HF mice consumed a greater amount during the light phase and less during the dark phase, yet their total amount (i.e. grams consumed, not

kcal consumed) of food intake over a 24 h period did not differ. This finding indicates that the high fat diet itself altered circadian behavior since the HF group was eating more during the time that mice typically rest. This alteration in eating behavior is consistent to what was found in male C57BL/6J mice [16].

In humans, obesity and circadian disruption, as imposed by shiftwork or sleep disruption, have been linked with elevated cortisol and an attenuation of rhythms, including in pregnant women, suggesting that both may induce physiological stress [33, 34]. We measured levels of hair corticosterone, the primary glucocorticoid in rodents, as a biomarker of chronic stress [10, 35, 36]. HF mice had elevated hair corticosterone, indicating a higher level of physiological stress. There was no relationship between hair corticosterone concentration and weight nor BMI of mice, indicating that diet, but not weight nor BMI was underlying the elevation in corticosterone. More studies are needed to understand if this is the high fat alone or changes in eating behavior patterns, with activity at normal times of rest increasing the corticosterone levels and the relationship of these changes to physiological stress.

Analysis of circadian secretion of corticosterone using fecal material found that CON mice had a higher amplitude for corticosterone concentration and exhibited an expected corticosterone circadian rhythm of secretion, with levels decreasing progressively throughout their active period (dark phase) and rising steadily during their rest to peak right before waking. The better fit of fecal weight and fecal corticosterone levels to two component cosine curves, suggest two periods of heightened evacuation. The first just after the onset of the dark phase, and the second at the transition between the dark and light phases. HF mice exhibited an elevated mesor, indicating higher basal levels of corticosterone, a lower amplitude, indicating an attenuation of rhythms, and shift in phase of corticosterone rhythm, where the peak corticosterone concentration was not at the onset of darkness like CON mice. The shift in corticosterone rhythms reflect the change in eating activity, with animals on a HF diet spending more time eating during the day (light-phase). The elevated levels suggest that high fat diet eating interacts with the hypothalamus-pituitary-adrenal (HPA) axis. Stress activates the HPA axis, resulting in corticosterone secretion [37]. Rodents on a high fat diet experience chronic stress, thus increasing activity of the HPA axis [38, 39]. The chronic exposure to elevated corticosterone levels increases the allostatic load of the animal. In humans, increased allostatic load is linked to poorer sleep, which disrupts circadian rhythms and increases the higher the risk for disease [40].

High fat diets disrupt circadian rhythms and impact the HPA axis, and the disruption of one system impacts other systems [41]. One system of particular interest to our work is the hypothalamic-pituitary-gonadal (HPG) axis. Stress has been shown to decrease female fertility by inhibiting GnRH, the hormone responsible for the cascade of female reproductive hormone release [20, 42]. Moreover, disrupted circadian rhythms and behavior due to HF diets may also lead to reproductive dysfunction, as circadian clocks regulate the timing of ovulation [14]. Wherein circadian disruption decreases female fertility [43].

There are several limitations to our study. The first is the use of ICR mice. Since ICR mice are an outbred strain, there is a high degree of variability between the mice, although this may better model responses of humans to nutritional environment. Secondly, for fecal corticosterone data and feed intake, the experimental unit was the cage, which housed between three to five mice per cage. This precluded analysis of individual animal fecal corticosterone and feed intake, which should be investigated further. Moreover, the fecal weight was not determined on a dry matter basis. Fecal material of mice is relatively dry, and others reported that there was no difference in fecal water content between mice fed a control versus high fat diet [44], supporting that the total amount of corticosterone in fecal material reported here was affected by diet.

## Conclusion

Feeding female ICR mice a high fat diet for four weeks led to significantly greater weight gain than mice on a control diet. Proceeding differences in weight were changes to diurnal eating patterns, wherein mice on a high fat diet consumed more food during the light phase when they should normally be resting. Alterations in eating patterns were related to a shift in the phase of circadian rhythms of fecal corticosterone and a decrease in amplitude with a concomitant rise in basal levels. These changes are reflective of disturbances in the HPA axis, and may be indicative of physiological stress with higher accumulation of hair corticosterone levels, which was independent of final weight of the mice. How these physiological changes may relate or underlie the impact of high fat diet on reproductive competence needs to be further investigated.

## Author Contributions

**Conceptualization:** Kelsey Teeple, Yu Han-Hallett, Theresa Casey.

**Data curation:** Kelsey Teeple, Prabha Rajput, Maria Gonzalez, Yu Han-Hallett, Esteban Fernández-Juricic, Theresa Casey.

**Formal analysis:** Kelsey Teeple, Maria Gonzalez, Esteban Fernández-Juricic, Theresa Casey.

**Investigation:** Kelsey Teeple, Prabha Rajput, Yu Han-Hallett, Theresa Casey.

**Methodology:** Kelsey Teeple, Prabha Rajput, Maria Gonzalez, Yu Han-Hallett, Esteban Fernández-Juricic, Theresa Casey.

**Project administration:** Theresa Casey.

**Resources:** Theresa Casey.

**Software:** Kelsey Teeple, Maria Gonzalez, Esteban Fernández-Juricic.

**Supervision:** Kelsey Teeple, Theresa Casey.

**Validation:** Yu Han-Hallett, Theresa Casey.

**Visualization:** Kelsey Teeple, Maria Gonzalez, Esteban Fernández-Juricic, Theresa Casey.

**Writing – original draft:** Kelsey Teeple.

**Writing – review & editing:** Prabha Rajput, Maria Gonzalez, Yu Han-Hallett, Esteban Fernández-Juricic, Theresa Casey.

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
