## [Decision Letter · Decision Letter 0]

10 Oct 2022

PONE-D-22-24766High fat diet induces obesity, alters eating behavior and disrupts corticosterone circadian rhythms in female ICR micePLOS ONE

Dear Dr. Casey,

Thank you for submitting your manuscript to PLOS ONE. After careful consideration, we feel that it has merit but does not fully meet PLOS ONE’s publication criteria as it currently stands. Therefore, we invite you to submit a revised version of the manuscript that addresses the points raised during the review process.

Please carefully add al the information the reviewers asked for.Fix normaöization for feces weight as requested.consider assessing stress more directly, or tone down conclusions in this respect.==============================

We look forward to receiving your revised manuscript.

Kind regards,

Henrik Oster, Ph.D.

Academic Editor

PLOS ONE

5. Please upload a copy of Supporting Information Figure/Table/etc. which you refer to in your text on page 9.

Additional Editor Comments :

The reviewers requested several clarifications regarding methodology and normalization, and I agree with them. Please add an experiment to directly assess stress in HFD animals, or tone done you interpretations with regard to tonic stress in these aninmals.

Reviewers' comments:

Reviewer's Responses to Questions

**Comments to the Author**

1. Is the manuscript technically sound, and do the data support the conclusions?

Reviewer #1: Yes

Reviewer #2: Yes

2. Has the statistical analysis been performed appropriately and rigorously? 

Reviewer #1: Yes

Reviewer #2: Yes

3. Have the authors made all data underlying the findings in their manuscript fully available?

Reviewer #1: Yes

Reviewer #2: Yes

4. Is the manuscript presented in an intelligible fashion and written in standard English?

Reviewer #1: Yes

Reviewer #2: Yes

5. Review Comments to the Author

Reviewer #1: Teeple et al. investigate the disruption of corticosterone circadian rhythm in obese female mice. They measured diurnal food intake and after 4wk of HFD they measured hair and fecal corticosterone, mouse weight gain, and body composition. The experiments were part of a larger study and were straightforward. They showed that HFD disturbs circadian rhythm, elevates fecal corticosterone, and changes circadian eating behavior that could be relevant during pregnancy. Although the study was well designed and well written there are some questions that need to be clarified:

45 what is specifically regulated reciprocally?

64 how exactly is cortisol changed diurnally?

105 The characteristics of the ICR mice strain are not clearly referenced-described.

117 what is the composition of the diet fat?

129 How is EcoMRA working, and what is measuring?

185 why corticosterone ELISA could not be used to analyze hair extracts?

193-195 Did you try to analyze food consumption vs food weight? Weight in Kcal is only a derivative of food weight, not a real variable; can you simplify your models? T

Figure explanations should not be embedded but to be together in Figure legend after Tables and References; It was hard to read them the way you show them.

Reviewer #2: This is a good paper to study the relationship between nutrition, circadian rhythms, and eating pattern. No major issues identified in experimental design. But some details need to be included in the revision. Additionally, the fecal weight needs to be corrected on a dry matter basis and on a per animal basis, if authors have not done so.

Line 22 and title

The word behavior for feeding or eating could represent the actual behaviors such as number of eating bout, length of one eating bout etc. Therefore, maybe diurnal eating pattern would be a better expression? --- Just a suggestion.

Line 33

“elevated basal corticosterone”

Line 34-35

This sentence may be reworded. Current form is hard to follow.

Line 115

Please have a unit following the SEM.

Was there a specific reason why the number of the mice in one cage not standardized?

Line 116

Can authors please list the ingredients of the two diets? Were both diets isonitrogenous?

Line 123

There seems to be dead animals in each cohort. Authors please provide details on mortality per treatment.

Line 193

Believe it should be “generalized linear mixed model”

Line 220

The fecal weight and total corticosterone data should be normalized by the number of animals per cage before analysis. Additionally, the fecal weight needs to corrected with dry matter, the same for corticosterone weight.

Table 2

Please include unit for the amount of fecal corticosterone. Again, the fecal weight and corticosterone weight need to normalized to number of animals per cage, and all weights need to be on a dry matter basis.

Line 296

Delete “not”

Line 308

If authors decide to use two component fitting for corticosterone concentrations, please leave the figure for two component fitting. It just doesn’t make sense to have one component in the figure.

Line 313-314

Can authors define amplitude?

Line 322-323

Authors please specify the gray line is for HF animals. Also please change black to solid.

Line 370

As indicated in following paragraph, the increased corticosterone deposition in hair could also be attributed to the increased basal concentration of corticosterone. Although authors emphasize the importance of cortisol as a stress marker, which may be right, it is still possible that HF diet will increase the basal secretion of corticosterone without stress induction. The other thing which is not clear is what type of chronic stress the HF animals were experiencing as authors indicated on line 389. Indeed, with higher basal level of corticosterone and loss of rhythms, it is also possible that the HPA axis is desensitized due to the negative feedback loop. A good way to test this would be perform a stress induction.

Figure 4

The quality of this figure is really poor. Please change to a high resolution one.

6. PLOS authors have the option to publish the peer review history of their article (what does this mean?). If published, this will include your full peer review and any attached files.

Reviewer #1: No

Reviewer #2: No

---

## [Author Response · Author response to Decision Letter 0]

21 Nov 2022

Response to Review

Dr. Oster and reviewers-

 Thank you so much for spending the time to review and provide feedback on our manuscript entitled “High fat diet induces obesity, alters eating pattern and disruption corticosterone circadian rhythms in female ICR mice” with the reference number PONE-D-22-24766 for consideration to be published in PLOS One.

We appreciate the questions and comments that were posed. We believe that in addressing these concerns, the quality of our manuscript has vastly improved and contributes to scientific rigor.

Please see the responses (RE) to the comments from the reviewers and the journal.

Best regard, Kelsey

RE: Manuscript has been formatted to comply with PLOS One’s requirements.

RE: Lines 138 – 142

RE: This project was funded by Purdue University as part of AgSEED Crossroads funding to support Indiana’s Agriculture and Rural Development and will be added during the submission process.

RE: https://figshare.com/s/7ad3a3b95ce66b1575df

5. Please upload a copy of Supporting Information Figure/Table/etc. which you refer to in your text on page 9.

 RE: This supporting information has been changed to reflect that the data is freely available at figshare, as well as the link to the data/R code.

RE: References were changed to Vancouver stye and URLs were added to websites, such as R packages. 

Additional Editor Comments:

The reviewers requested several clarifications regarding methodology and normalization, and I agree with them. Please add an experiment to directly assess stress in HFD animals, or tone done you interpretations with regard to tonic stress in these aninmals.

Reviewers' comments:

Reviewer's Responses to Questions

Comments to the Author

Reviewer #1: Teeple et al. investigate the disruption of corticosterone circadian rhythm in obese female mice. They measured diurnal food intake and after 4wk of HFD they measured hair and fecal corticosterone, mouse weight gain, and body composition. The experiments were part of a larger study and were straightforward. They showed that HFD disturbs circadian rhythm, elevates fecal corticosterone, and changes circadian eating behavior that could be relevant during pregnancy. Although the study was well designed and well written there are some questions that need to be clarified:

1. 45 what is specifically regulated reciprocally? 

RE: clarified (line 49)

2. 64 how exactly is cortisol changed diurnally? 

RE: Peaks before waking and dwindles throughout the day and it’s at the lowest right before sleep (lines 59-61)

3. 105 The characteristics of the ICR mice strain are not clearly referenced-described. 

RE: Added characters in line 98-99

4. 117 what is the composition of the diet fat? 

RE: added line 124-128

5. 129 How is EcoMRA working, and what is measuring? 

RE: EchoMRI measures the differences in times of hydrogen proton spins between soft and firm tissues. Radio pulses result in proton spins that emit radio signals that are analyzed by the EchoMRI. To differentiate between fat, lean, and free water, a specifically composed radio pulse signal is used. – added line 140-143

6. 185 why corticosterone ELISA could not be used to analyze hair extracts? 

RE: The Arbor Assay corticosterone ELISA can be used for hair extractions as well. LC-MS/MS was initially used to measure hair corticosterone. We then tested whether LC-MS/MS could be used to measure fecal corticosterone. We found that the metabolites within the fecal matrix interfered with analysis. The low levels of corticosterone in fecal material and the fact that other compounds co-eluting, made it difficult to have precise quantification. (Line 192-194)

7. 193-195 Did you try to analyze food consumption vs food weight? Weight in Kcal is only a derivative of food weight, not a real variable; can you simplify your models? 

RE: No, we did not. We only measured feed intake in grams and converted to kcal. Perhaps in another study this could be done.

8. Figure explanations should not be embedded but to be together in Figure legend after Tables and References; It was hard to read them the way you show them.

RE: This is the style that PLOS One uses for submission.

Reviewer #2: This is a good paper to study the relationship between nutrition, circadian rhythms, and eating pattern. No major issues identified in experimental design. But some details need to be included in the revision. Additionally, the fecal weight needs to be corrected on a dry matter basis and on a per animal basis, if authors have not done so.

1. Line 22 and title

The word behavior for feeding or eating could represent the actual behaviors such as number of eating bout, length of one eating bout etc. Therefore, maybe diurnal eating pattern would be a better expression? --- Just a suggestion. 

RE: edited as suggested.

2. Line 33

“elevated basal corticosterone”

RE: Changed

3. Line 34-35

This sentence may be reworded. Current form is hard to follow.

RE: Revised sentences

4. Line 115

Please have a unit following the SEM.

RE: Added units following SEM

5. Was there a specific reason why the number of the mice in one cage not standardized?

RE: Rodent housing is on a per cage cost, so standardizing to an even number would have increased the number of cages per treatment, therefore increasing the overall cost. We can house 5 mice per cage, so we housed 5 mice in all cages. In cohort 1, there was one CON cage that had 3 mice, and there was one HF cage that had 4 mice. In cohort 2, there was again one CON cage that had 3 mice.

6. Line 116

Can authors please list the ingredients of the two diets? Were both diets isonitrogenous?

RE: Added that diets were isonitrogenous as well as referenced where the dietary composition was previously published.

7. Line 123

There seems to be dead animals in each cohort. Authors please provide details on mortality per treatment. 

RE: Thank you for highlighting this, we had removed six animals that were likely morbid and lost weight throughout the course of the study. However, these animals could not be removed from the feed intake and fecal corticosterone date, so we analyzed weight gain with and without them. We analyzed growth rates, BMI, MRI, and hair corticosterone with and without the morbid mice, and regardless of their inclusion or not, HF had the same effects on these outcomes.

8. Line 193

Believe it should be “generalized linear mixed model” 

RE: Corrected to generalized linear mixed model

9. Line 220

The fecal weight and total corticosterone data should be normalized by the number of animals per cage before analysis. Additionally, the fecal weight needs to corrected with dry matter, the same for corticosterone weight.

RE: Thank you for pointing this out. We have expressed all data on a per mouse basis. However, the total fecal weight was not determined on a dry matter basis, approximately 0.2 g was weighed then dried. Although mouse fecal material is quite dry, there is the potential that difference in abundance could be due to differences in water content. Although, , others found that water content of feces were the same in mice fed a control versus high fat diets (Mukai et al., 2020). We added this as a limitation to our study.

10. Table 2

Please include unit for the amount of fecal corticosterone. Again, the fecal weight and corticosterone weight need to normalized to number of animals per cage, and all weights need to be on a dry matter basis.

RE: This has been fixed to a per mouse basis.

11. Line 296

Delete “not”

RE: Deleted “not” in line 308

12. Line 308

If authors decide to use two component fitting for corticosterone concentrations, please leave the figure for two component fitting. It just doesn’t make sense to have one component in the figure.

RE: The single component is removed.

13. Line 313-314

Can authors define amplitude? 

RE: Amplitude was defined in line 324

14. Line 322-323

Authors please specify the gray line is for HF animals. Also please change black to solid.

RE: Grey lines were specified in the Fig. 4 description to represent the HF animals

15. Line 370

As indicated in following paragraph, the increased corticosterone deposition in hair could also be attributed to the increased basal concentration of corticosterone. Although authors emphasize the importance of cortisol as a stress marker, which may be right, it is still possible that HF diet will increase the basal secretion of corticosterone without stress induction. The other thing which is not clear is what type of chronic stress the HF animals were experiencing as authors indicated on line 389. Indeed, with higher basal level of corticosterone and loss of rhythms, it is also possible that the HPA axis is desensitized due to the negative feedback loop. A good way to test this would be perform a stress induction.

RE: Thank you for the suggestion and it is a good idea to perform and should be considered in the future to further understand how elevated corticosterone is impacting the HPA. Investigating a stress induction test is beyond the scope of the current study.

16. Figure 4

The quality of this figure is really poor. Please change to a high resolution one.

RE: The figure was saved as a higher quality.

Mukai, R., Handa, O., Naito, Y., Takayama, S., Suyama, Y., Ushiroda, C., Majima, A., Hirai, Y., Mizushima, K., Okayama, T., Katada, K., Kamada, K., Uchiyama, K., Ishikawa, T., Takagi, T., Itoh, Y., 2020. High-Fat Diet Causes Constipation in Mice via Decreasing Colonic Mucus. Digestive Diseases and Sciences 65 (8), 2246-2253.

---

## [Editor Report · Decision Letter 1]

2 Dec 2022

High fat diet induces obesity, alters eating pattern and disrupts corticosterone circadian rhythms in female ICR mice

PONE-D-22-24766R1

Dear Dr. Casey,

We’re pleased to inform you that your manuscript has been judged scientifically suitable for publication and will be formally accepted for publication once it meets all outstanding technical requirements.

Kind regards,

Henrik Oster, Ph.D.

Academic Editor

PLOS ONE

Additional Editor Comments (optional):

Congrats on this nice paper.
---

## [Editor Report · Acceptance letter]

10 Jan 2023

PONE-D-22-24766R1 

High fat diet induces obesity, alters eating pattern and disrupts corticosterone circadian rhythms in female ICR mice 

Dear Dr. Casey:

I'm pleased to inform you that your manuscript has been deemed suitable for publication in PLOS ONE. Congratulations! Your manuscript is now with our production department. 

Kind regards, 

on behalf of

Prof. Henrik Oster 

Academic Editor

PLOS ONE